# Robotic Mediastinal Tumor Resections: Position and Port Placement

**DOI:** 10.3390/jpm12081195

**Published:** 2022-07-22

**Authors:** Mikio Okazaki, Kazuhiko Shien, Ken Suzawa, Seiichiro Sugimoto, Shinichi Toyooka

**Affiliations:** Department of Nephrology, Rheumatology, Endocrinology and Metabolism, Graduate School of Medicine, Dentistry and Pharmaceutical Sciences, Okayama University, Okayama 700-8530, Japan; k.shien@okayama-u.ac.jp (K.S.); ksuzawa@okayama-u.ac.jp (K.S.); sugimo-s@cc.okayama-u.ac.jp (S.S.); toyooka@md.okayama-u.ac.jp (S.T.)

**Keywords:** robot, robot-assisted thoracic surgery, mediastinal tumor, thymectomy, port placement

## Abstract

This study aimed to determine the optimal position and port placement during robotic resection for various mediastinal tumors. For anterior mediastinal tumors, total or extended thymectomy is commonly performed in the supine position using the lateral or subxiphoid approach. Although it is unclear which approach is better during robotic thymectomy, technical advantages of subxiphoid approach are beneficial for patients with myasthenia who require extended thymectomy. Partial thymectomy is performed in the supine position using a lateral approach. Superior, middle, and posterior mediastinal tumors are resected in the decubitus position using the lateral approach, whereas dumbbell tumor resection, which requires a posterior approach, can be performed in the prone position. The position and port placement should be chosen depending on the size, location, and aggressiveness of the tumor. In this study, we describe how to choose which of these different robotic approaches can be used based on our experience and previous reports.

## 1. Introduction

Mediastinal masses comprise a heterogeneous group of tumors including thymoma, thymic cancer, germ cell tumors, neurogenic tumors, various cysts, lymphoid tissues, and ectopic parathyroid or thyroid tumors. Furthermore, the localization of mediastinal tumors includes the anterior, superior, and posterior mediastinum. Therefore, the surgical procedure and position should be chosen based on the nature and localization of the tumor.

Compared with conventional median sternotomy and lateral thoracotomy, video-assisted thoracic surgery (VATS) for mediastinal tumors has become much less invasive recently. Additionally, it has been reported that the oncological outcomes of open and VATS thymectomies are comparable at mid-term and long-term follow-up evaluations [1,2,3]. Although there are patients that require thoracotomy because of the great vessel or chest wall invasion, it has become common to determine whether the tumor is resectable using a minimally invasive approach.

The first case of robotic resection of a mediastinal tumor was reported by Yoshino et al. [4]. Thereafter, the use of robot-assisted thoracic surgery (RATS) for the treatment of mediastinal tumors has been widely reported. However, few studies have directly compared RATS and VATS for mediastinal tumors. Alvaldo et al. examined 856 patients who underwent VATS or RATS for mediastinal tumors and found that RATS had fewer adverse events than VATS even with tumors larger than 4 cm [5]. Shen et al. also reported that RATS thymectomy was associated with less intraoperative bleeding, lower drainage volumes, fewer postoperative pleural drainage days, fewer postoperative complications, and shorter postoperative hospital stays than VATS thymectomy [6]. Therefore, RATS may be the preferred approach for mediastinal tumors when minimally invasive surgery is indicated.

Although various RATS approaches can be chosen based on the nature and localization of mediastinal tumors, few reports have summarized them. This study aimed to determine the optimal position and port placement for each mediastinal tumor during robotic resection.

## 2. Anterior Mediastinal Tumor

### 2.1. Total and Extended Thymectomy

RATS thymectomy is a safe and feasible procedure for anterior mediastinal tumors and myasthenia gravis (MG) [6,7,8], and it is commonly performed using the lateral approach. Recently, the subxiphoid approach has been reported as an alternative to the lateral approach for RATS thymectomy [9,10,11,12,13].

The lateral approach is usually performed via the right or left side; however, the bilateral approach may be used for extended thymectomy in patients with MG. Figure 1 shows the typical port placement for the three-arm lateral approach using the da Vinci Xi surgical system (Intuitive Surgical, Sunnyvale, CA, USA). 

The patient was placed in the supine position and moved to the edge of the operative side of the table (Figure 1A). The chest was lifted up by a pillow under the back. The ipsilateral forearm was fixed on an additional handstand at a lower position than the bed to provide a wide working space for robotic arms (Figure 1B). The contralateral forearm was fixed on the bed away from the chest so that contralateral approach could be used if necessary. The camera port was inserted at the fifth intercostal space on the mid-axillary line, and CO_2_ insufflation was performed (Figure 1C). Two additional trocars were inserted in the third intercostal space on the mid-axillary line and in the sixth intercostal space on the anterior axillary line. An assistant suction device can be used between the camera port and left arm port if necessary. A right-side approach has a wider field of view and larger working space [14,15,16,17,18,19], whereas a left-side approach potentially accesses more thymic tissue and allows easier visualization of the contralateral phrenic nerve [20,21,22,23,24,25,26]. Although it is controversial whether a left-side or right-side approach is better, several recent studies have reported using the left-side approach for extended thymectomy in patients with MG because it provides enhanced visualization and reduces the probability of phrenic nerve injury [21,25,27,28]. Wilshire et al. reviewed 123 patients who underwent minimally invasive thymectomy for MG and reported that left-side thymectomy was associated with shorter operative times and the potential for superior medium-term symptomatic outcomes compared to right-side thymectomy [25]. However, it should be noted that unfamiliar and inadvertent port placement can cause cardiac injury during left-side thymectomy [21,29]. For patients with an anterior mediastinal mass, we usually use an ipsilateral approach to the tumor location because tumors can be seen clearly, thereby reducing the risk of dissemination by grasping or breaking the tumor inadvertently. Furthermore, we use the supine position instead of the semi-lateral position for the lateral approach because the contralateral side can be used when the contralateral phrenic nerve cannot be seen or additional tissue resection of the contralateral side is required. Additionally, median sternotomy, which may be urgently required because of massive bleeding, is easier to perform in the supine position than in the semi-lateral position. 

Figure 2 shows the typical port placement for the three-arm subxiphoid approach.

The patient was placed in the supine position and the chest was lifted up by a pillow under the back. The both arms were abducted so as not to interfere with the movement of the robotic arms, and the legs were not opened. The surgeon stood on the right side of the patient, and the endoscopist stood on the left side. A 3 cm vertical incision was made below the xiphoid process. After dividing the linea alba, the attachment of the rectus abdominis was dissected away from the xiphoid process, and the posterior surface of the sternum was blindly detached using the finger. A 4-channel glove port was inserted, and CO_2_ insufflation was performed. The bilateral mediastinal pleura was incised using electrocautery, and the surrounding area was further dissected until there was sufficient space between the sternum and pericardium. Two additional robotic ports were inserted bilaterally at the sixth intercostal space along the anterior axillary line. One of the subxiphoid ports was used as an assist port. If the 4-arm subxiphoid approach is performed, then an additional robotic port is inserted at the right sixth intercostal space along the mid-axillary line. Most reports of subxiphoid approach have been performed with the legs open [29,30], but these preparations can be performed without opening the legs.

### 2.2. Lateral Versus Subxiphoid Approach

Although both lateral and subxiphoid approaches have been reported [7,8,9,10,11,12,14,28,31,32,33,34], it is unclear which approach is better during robotic thymectomy. We compared the surgical outcomes and factors contributing to low surgical invasiveness associated with these approaches. Between December 2018 and March 2022, 29 patients underwent robotic total or extended thymectomy for anterior mediastinal tumors or MG at our institution. Patients were divided into the lateral group (*n* = 13) or subxiphoid group (*n* = 16). Operative time and console time were not different between groups; however, the operative time minus the console time was significantly longer in the subxiphoid group than in the lateral group (Table 1). 

This is because it takes time to make a subxiphoid incision, dissect the surrounding tissue, create a bilateral opening of the pleura, and perform port placement before the robotic procedure. However, Park et al. reported that operative time of the subxiphoid approach was significantly shorter than that of the lateral approach [34]. Intraoperative bleeding, intraoperative complications, and postoperative complications were not significantly different between groups. The white blood cell count and C-reactive protein level collected on postoperative day 5 were not significantly different between groups. Postoperative pain was evaluated using a numerical rating scale on postoperative day 5; the numerical rating scale was not significantly different between groups. On the other hands, Park et al. reported that the subxiphoid approach was more significantly associated with lower pain score than the lateral approach [34]. The maximum incision size was significantly greater in the subxiphoid group than in the lateral group. The subxiphoid approach requires a 3 cm incision in the subxiphoid area. In contrast, the lateral approach requires only small incisions for 8 mm ports during robotic procedures, and one of the wounds is extended as long as necessary to remove the tumor. However, the subxiphoid approach may be preferred for tumors larger than 3 cm in diameter because it is easier to extract these tumors from the subxiphoid incision than from the intercostal incision. 

Suda et al. first reported robotic thymectomy using the subxiphoid approach and noted that this technique provides a good surgical view of the neck region and facilitates verification of the phrenic nerve [30]. Although surgical outcomes were not significantly different between the lateral and subxiphoid approaches during the present study, Hashimoto et al. demonstrated that the subxiphoid approach was more advantageous than the lateral approach for visualization of the phrenic nerve and dissection of the superior poles during RATS thymectomy [35]. These technical advantages are especially beneficial for patients with myasthenia who require extended thymectomy. Furthermore, the subxiphoid approach is recommended for patients with tumors located above the innominate vein because the subxiphoid approach provides a better operative view in the upper mediastinal region than the lateral approach. However, both approaches should be considered for total thymectomy because it can be performed even if the contralateral phrenic nerve is not visible. 

### 2.3. Partial Thymectomy 

Partial thymectomy is performed for anterior mediastinal tumors, such as thymic cysts, pericardial cysts, ectopic thyroid or parathyroid tumors, and mature teratomas. The lateral approach is usually recommended for partial thymectomy because it is not necessary to check the contralateral phrenic nerve, cysts and small tumors involving parts of the thymus gland can usually be removed through a small incision, and it is more advantageous in terms of cosmetic outcomes than the subxiphoid approach.

## 3. Posterior Mediastinal Tumor

### 3.1. Posterior Tumor Resection

Most posterior mediastinal tumors are benign neurogenic tumors, such as schwannomas, neurofibromas, and ganglioneuromas. Patients are placed in the decubitus position [36] except when they have dumbbell tumors. Port placement depends on the tumor location. We used three robotic ports with linear port placement according to the position of the tumor. Figure 3 shows a patient with a right posterior mediastinal tumor at the Th8–9 level (Figure 3A,B). 

A camera port was inserted at the sixth intercostal space in the mid-axillary line, and CO_2_ insufflation was performed. Two additional trocars were inserted in the fourth intercostal space on the anterior axillary line and in the eighth intercostal space on the posterior axillary line (Figure 3C). An assistant port can be placed in the seventh intercostal space on the anterior axillary line if necessary. The tumor was well-visualized and resected without complications. When using Si for inferior mediastinal tumors, the patient cart should roll from the caudal side and dock on the backside [36,37]. 

### 3.2. Dumbbell Tumor Resection

Resection of Eden type 2 or 3 dumbbell thoracic tumors requires both the anterior approach and posterior approach. The standard procedure for dumbbell tumors is a combination of laminectomy in the prone position and VATS resection in the lateral recumbent position. However, we previously reported a combined posterior approach and robot-assisted resection for dumbbell tumors in the prone position [38]. Magnetic resonance imaging revealed a dumbbell-shaped right foraminal and paravertebral tumor (Figure 4A). 

After laminectomy and dissection of the dorsal part of the tumor using a posterior incision (Figure 4B), the camera port was placed in the seventh intercostal space on the posterior axillary line. Two additional ports were inserted in the fifth intercostal space on the mid-axillary line and the ninth intercostal space on the back (Figure 4C). An assist port was placed in the ninth intercostal space on the mid-axillary line, and CO_2_ insufflation was performed. The tumor was completely resected in the thoracic cavity using a robotic approach. The most important advantage of this approach is that repositioning is not necessary during the procedure, and we can choose back or robotic approaches at any time. The back wound was used as an outlet for the tumor, and hemostasis was confirmed using the back incision after robotic tumor resection. Flexible wrist mechanisms allow for easy and quick resection of dumbbell tumors to avoid repositioning and reduce the procedure time. 

## 4. Superior Mediastinal Tumor

Most superior mediastinal tumors are benign neurogenic tumors, such as schwannomas, except for malignant bulky tumors. However, a narrow working space results in surgical difficulty during the resection of sulcal tumors. Moreover, the procedure involves the risk of injury to vital anatomical structures, such as the subclavian artery and recurrent laryngeal nerve around the tumors. Robots may be a solution to these problems because they are suitable for surgery in a narrow space with three-dimensional vision and flexible wrist instruments. Wang et al. reported 15 cases of robotic resection of superior sulcus neurogenic tumors without conversion and suggested that RATS might be a promising alternative modality for the resection of superior sulcus tumors [39]. The patient was placed in the lateral decubitus position, and three robotic ports with or without an assistant port were used [39,40]. Figure 5 shows a patient with a superior sulcus schwannoma.

The camera port was inserted in the sixth intercostal space along the mid-axillary line, and CO_2_ insufflation was performed. Two additional trocars were inserted in the fourth intercostal space on the anterior axillary line and in the seventh intercostal space on the posterior axillary line (Figure 5C). An assistant port was placed in the sixth intercostal space on the anterior axillary line if necessary. The tumor was enucleated without electrocautery or bipolar cautery, and there were no perioperative complications, such as Horner’s syndrome or recurrent laryngeal nerve palsy.

## 5. Middle Mediastinal Tumor

Bronchogenic and pericardial cysts may occur in the middle mediastinum and are indications for robotic surgery. The patient was placed in the lateral decubitus position, and three robotic ports with or without an assistant port were used. Port placement depends on the tumor location. We used three robotic ports with linear port placement according to the position of the tumor, similar to posterior tumor resection.

## 6. Literature Summary and Preferred Position and Approach for RATS Mediastinal Tumor Resection

The literature reviewed in the study are summarized in Table 2. Reports of anterior mediastinal tumors with more than 50 cases and descriptions of operative approach are listed. Reports of superior or posterior mediastinal tumors with more than 15 cases and descriptions of operative approach are listed. The preferred position and approach according to tumor location and surgical procedure are summarized in Table 3. These will allow for RATS for patients with mediastinal tumors to be performed more safely and easily. All cases considered resectable by VATS can be resected by RATS. Although large tumors less than around 10 cm are also resectable by robotic approach, malignant tumors invading the great vessels or chest wall are usually not indication for RATS but for open thoracotomy [26,34]. However, there are some reports of RATS thymectomy with combined resection of the brachiocephalic vein or superior vena cava [12,34]. RATS may further expand its indication for mediastinal tumors as robots evolve.

## 7. Conclusions

RATS resection of various mediastinal tumors can be performed as previously described. However, RATS requires more safety measures than VATS because the conversion to open thoracotomy from RATS is more time-consuming than that from VATS in emergency cases. Although some approaches are controversial, it may be important to choose an approach that is comfortable for all surgeons performing RATS.

## Figures and Tables

**Figure 1 jpm-12-01195-f001:**
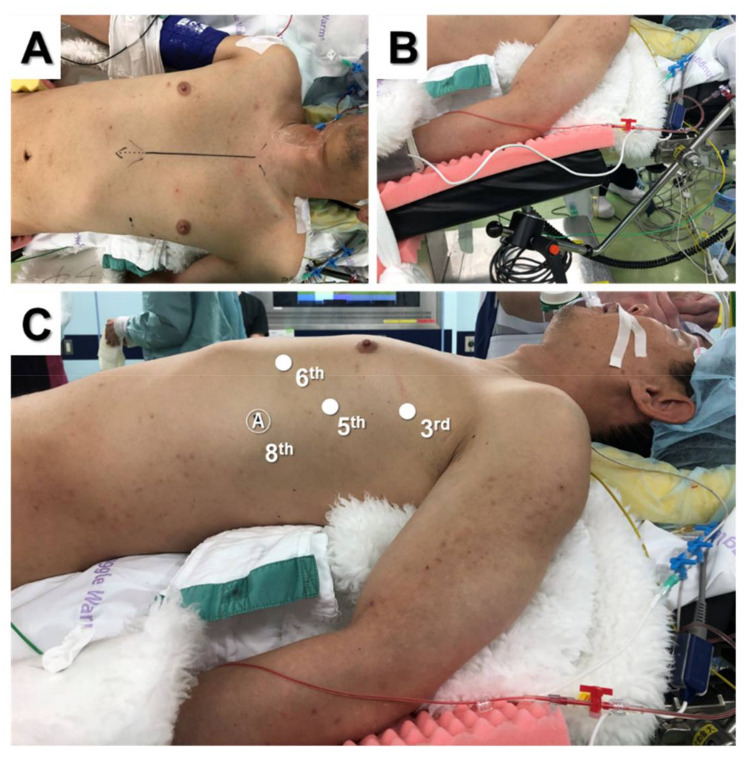
Supine position for three-arm robotic thymectomy using the lateral approach (**A**). The ipsilateral forearm was fixed at a lower position than the bed (**B**). Port placement (**C**).

**Figure 2 jpm-12-01195-f002:**
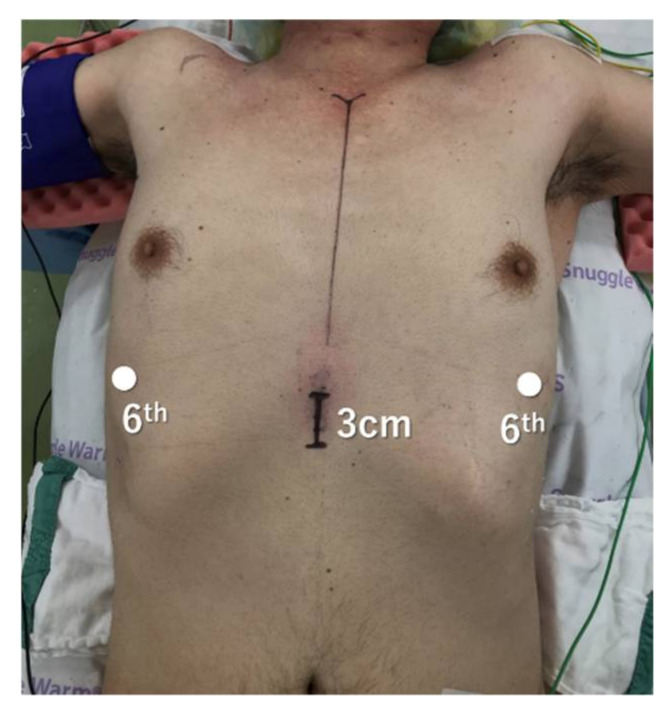
Port placement for three-arm robotic thymectomy using the subxiphoid approach.

**Figure 3 jpm-12-01195-f003:**
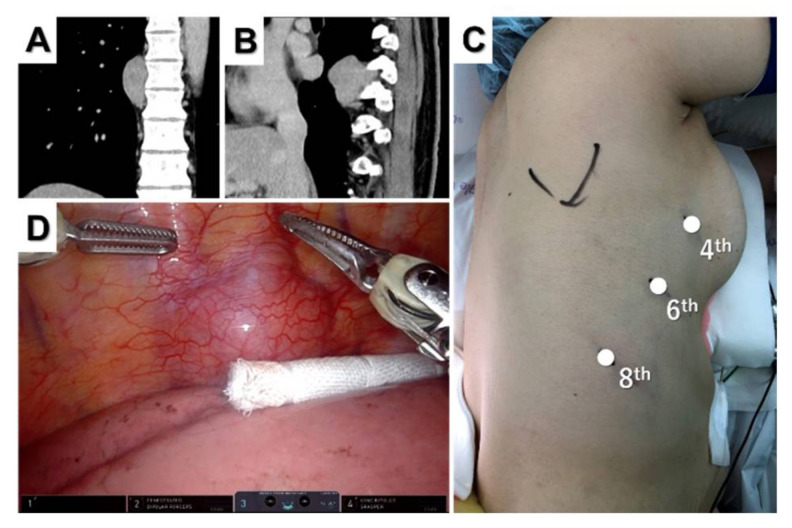
Posterior mediastinal tumor. Coronal (**A**) and sagittal (**B**) computed tomography views of the tumor located at the Th8–9 level. Port placement for three-arm robotic resection of a posterior mediastinal tumor (**C**) and intraoperative thoracoscopic view of the tumor (**D**).

**Figure 4 jpm-12-01195-f004:**
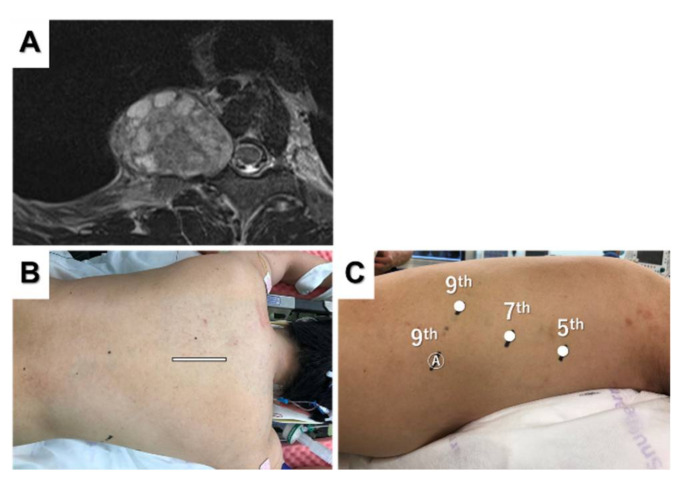
Magnetic resonance image showing a posterior mediastinal dumbbell tumor (**A**). The posterior incision (**B**) and port placement for robotic tumor resection (**C**) are shown.

**Figure 5 jpm-12-01195-f005:**
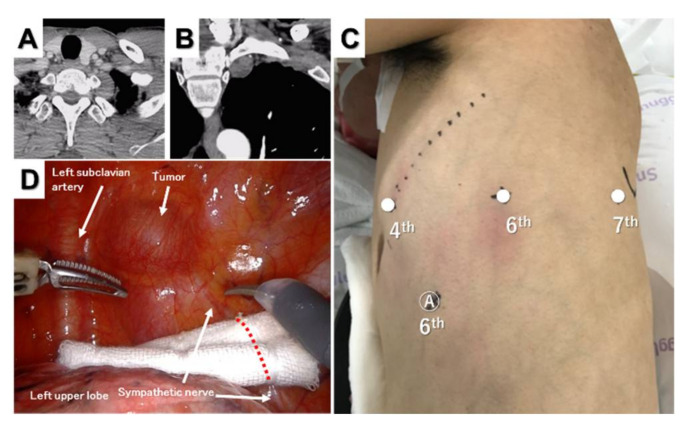
Superior sulcus tumor. Horizontal (**A**) and sagittal (**B**) computed tomography views of the tumor. Port placement for three-arm robotic resection of a superior sulcus tumor (**C**) and intraoperative thoracoscopic view of the tumor (**D**).

**Table 1 jpm-12-01195-t001:** Experience of robotic thymectomy using the lateral approach and subxiphoid approach.

	Lateral	Subxiphoid	*p*-Value
	(*n* = 13)	(*n* = 16)	
Patient characteristics			
Age (years old)	61.9 ± 14.8	59.1 ± 13.7	0.6023
Gender (male/female)	5/8	8/8	0.5344
Tumor size	3.1 ± 1.9	3.5 ± 1.9	0.6623
Procedure			0.0788
Extended thymectomy	2	7	
Total thymectomy	11	9	
Perioperative outcomes			
Operative time (min)	180.4 ± 61.6	185.6 ± 49.5	0.8011
Console time (min)	123.2 ± 60.0	110.8 ± 44.6	0.5280
Operative time–Console time (min)	57.2 ± 8.9	74.8 ± 10.5	<0.0001
Intraoperative bleeding (mL)	21.9 ± 23.6	14.4 ± 17.5	0.3312
Intraoperative complications	0 (0%)	0 (0%)	-
Postoperative complications	1 (7.7%)	0 (0%)	0.2589
Maximum size of incision (cm)	2.3 ± 0.6	3.0 ± 0.0	0.0001
WBC POD5 (×10^3^/mm^3^)	5.72 ± 1.01	6.19 ± 1.98	0.4555
CRP POD5 (mg/dL)	2.25 ± 2.03	3.59 ± 1.92	0.0799
NRS POD5	1.0 ± 0.6	0.8 ± 1.0	0.5157

**Table 2 jpm-12-01195-t002:** Literature summary of robotic mediastinal tumor resections including large case series and descriptions of approach.

Tumor Location	Author	Country	Year	Total	MG	Thymoma	Approach	Position	Ports	Conversion (%)
Anterior	Rückert	Germany	2008	106	95	12	Left	Semi-supine	3	1.1
Anterior	Freeman	USA	2011	75	75	0	Left	Supine	3	1.1
Anterior	Schneiter	Switzerland	2013	58	25	20	Left	Semi-supine	3	0.0
Anterior	Marulli	Italy	2013	79	45	79	Left	Semi-supine	3	1.3
Anterior	Jun	China	2014	55	n.a.	21	Right (>Left)	Supine	4	0.0
Anterior	Qian	China	2017	51	4	51	Right (>Left)	Semi-supine	4	0.0
Anterior	Kang	Korea	2020	110	18	67	Subxiphoid	Supine	3	0.9
Anterior	Li	China	2020	60	4	55	Right	Supine	3	n.a.
Anterior	Azenha	Switzerland	2021	81	n.a.	34	Left (>Right)	n.a.	3	0.0
Anterior	Marcuse	Netherlands	2021	130	89	130	Right (>Left)	Supine	3	7.7
Anterior	Romano	Italy	2021	53	34	53	Left (>Right)	Supine	3	1.9
Anterior	Wilshire	USA	2021	123	26	123	Left (>Right)	Semi-supine	3	n.a.
Anterior	Kang	Korea	2021	158	27	132	Subxiphiod/Left/Right	Supine	3	1.3
Anterior	Kumar	India	2021	111	68	111	Left/Right	Supine	3	6.3
Anterior	Geraci	USA	2021	84	n.a.	84	Right (>Left)	Semi-supine	3	2.3
Anterior	Zhang	China	2022	87	87	53	Subxiphoid	Supine	4	0.0
Anterior	Bongiolatti	Italy	2022	54	13	54	Left (>Right)	Semi-supine	3	11.1
Anterior	Park	Korea	2022	389	44	198	Subxiphiod/Left (>Right)	Supine	3	0.5
Superior	Wang	China	2021	15	-	-	Lateral	Lateral decubitus	3	0.0
Posterior	Cerfolio	USA	2012	75	-	-	Lateral	Lateral decubitus	4	1.3
Posterior	Li	China	2020	58	-	-	Lateral	Lateral decubitus	3	3.4

**Table 3 jpm-12-01195-t003:** Preferred position and approach for RATS mediastinal tumor resection according to tumor location and surgical procedure.

Tumor Location	Procedure	Preferred Position	Preferred Approach
Anterior mediastinum	Extended thymectomy	Supine	Subxiphoid
	Total thymectomy	Supine	Subxiphoid or lateral
	Partial thymectomy	Supine	Lateral
Superior mediastinum	Tumor resection	Lateral decubitus	Lateral
Middle mediastinum	Tumor resection	Lateral decubitus	Lateral
Posterior mediastinum	Tumor resection	Lateral decubitus	Lateral
	Dumbbell tumor resection	Prone	Lateral

## Data Availability

The data presented in this study are available on request from the corresponding author.

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
