# Peer review of "Robotic Mediastinal Tumor Resections: Position and Port Placement"

_jpm, 2022, doi:10.3390/jpm12081195_

Round 1
Reviewer 1 Report
very interesting and well prepared article
Author Response
We thank the reviewer for the time and effort to help us to improve the manuscript.
Reviewer 2 Report
The authors brilliantly reviewed the real-world evidence of robotic mediastinal tumor resection regarding port placement strategies, and concluded that robotic surgeons may choose their port placement strategy based on personal experience and previous literature.
The following are my comments.
1. As you presented your own data of RATS regarding different approach (lateral vs subxiphoid) in 29 patients undergoing anterior mediastinal tumor resection from Dec 2018 to Mar 2022. The case numbers was small and you may explain the reasons why case accumulation is slow (was it associated with COVID pandemic?). I would suggest you use the term “experience” rather than "comparison" to demonstrate your findings.
Further, the comparison from these two arms would lead to underpowered result.
While interpreting the data within the paragraph (line 122 to line 136), you should carefully pay attention to it instead of jumping to a firm conclusion. More relevant studies has been published in the literature and you are encouraged to include those for comparisons.
2. You should clearly describe the patient position regarding whether neck extension, upper chest extension, or arm position. RATS surgeons will be happy to know details about your expertise preoperative preparation on this issue (Such as Fig1 and Fig 2). A more detailed picture of the patient position would be recommended to add in the paper.
3. Regarding Fig 3 and Fig 4, you demonstrated the feasibility of RATS mediastinal tumor resection for such lesions. However, the patient selection is unclear, such as tumor size or association with adjacent anatomic structure (tightly adhesion or invasion?).
Moreover, as you mentioned that “RATS requires more safety measures than VATS because the conversion to open thoracotomy from RATS is more time-consuming than that from VATS in emergency cases”.
What is your indication and contraindication of RATS for mediastinal tumor resection? And if not RATS, when would conventional VATS is preferred? It should be concisely and clearly defined.
Author Response
The authors brilliantly reviewed the real-world evidence of robotic mediastinal tumor resection regarding port placement strategies, and concluded that robotic surgeons may choose their port placement strategy based on personal experience and previous literature.
The following are my comments.
1. As you presented your own data of RATS regarding different approach (lateral vs subxiphoid) in 29 patients undergoing anterior mediastinal tumor resection from Dec 2018 to Mar 2022. The case numbers was small and you may explain the reasons why case accumulation is slow (was it associated with COVID pandemic?). I would suggest you use the term “experience” rather than "comparison" to demonstrate your findings.
Further, the comparison from these two arms would lead to underpowered result.
While interpreting the data within the paragraph (line 122 to line 136), you should carefully pay attention to it instead of jumping to a firm conclusion. More relevant studies has been published in the literature and you are encouraged to include those for comparisons.
Response: We appreciated the reviewer’s comment. The number of cases at our institution is not very large, and the number is the same as in previous years. Therefore, we used “experience” rather than “comparison” (line 121).
We were not aware of the literature which the reviewer pointed out because it had just been published. We included the literature and changed sentences (line 126-127, line 132-134).
2. You should clearly describe the patient position regarding whether neck extension, upper chest extension, or arm position. RATS surgeons will be happy to know details about your expertise preoperative preparation on this issue (Such as Fig1 and Fig 2). A more detailed picture of the patient position would be recommended to add in the paper.
Response: We appreciated the reviewer’s comment. We changed Figure 1 and added sentences (line 64-70) to make patient’s position clear.
3. Regarding Fig 3 and Fig 4, you demonstrated the feasibility of RATS mediastinal tumor resection for such lesions. However, the patient selection is unclear, such as tumor size or association with adjacent anatomic structure (tightly adhesion or invasion?).
Moreover, as you mentioned that “RATS requires more safety measures than VATS because the conversion to open thoracotomy from RATS is more time-consuming than that from VATS in emergency cases”.
What is your indication and contraindication of RATS for mediastinal tumor resection? And if not RATS, when would conventional VATS is preferred? It should be concisely and clearly defined.
Response: We appreciated the reviewer’s comment. We added the sentence to describe the indication of dumbbell tumor resection in the prone position (line 181). Furthermore, we added sentences describing the indications of RATS mediastinal tumor resection (line 240-245).
We thank the reviewer for the time and effort to help us to improve the manuscript.
Reviewer 3 Report
Dear authors,
The authors have carried out ta review article tittled “Robotic mediastinal tumor resections: Position and port placement”. The aim of this study was to to determine the optimal position and port placement for each mediastinal tumor during robotic resection.
The authors have conducted a comprehensive, rigorous, and scientifically correct review of the subject.
Nevertheles, some considerations need to be taken into account:
§ In the manuscript, only the personal experience of the authors with respect to the anterior mediastinal tumor is collected in Table 1. The personal experience in the approach to superior, posterior or middle mediastinal tumors is not stated in the text. What is the explanation for this fact?
§ It would be desirable to identify in the manuscript the most representative studies on which the review has been based and reflect them in the legend of the text. The use of an accessory table would be appreciated
§ It is not usual to include a summary table with the personal recommendations of the authors in the conclusions of a review. Please consider its inclusion in a different section of the manuscript.
§ References are updated.
Kind regards
Author Response
Dear authors,
The authors have carried out ta review article tittled “Robotic mediastinal tumor resections: Position and port placement”. The aim of this study was to to determine the optimal position and port placement for each mediastinal tumor during robotic resection.
The authors have conducted a comprehensive, rigorous, and scientifically correct review of the subject.
Nevertheles, some considerations need to be taken into account:
- In the manuscript, only the personal experience of the authors with respect to the anterior mediastinal tumor is collected in Table 1. The personal experience in the approach to superior, posterior or middle mediastinal tumors is not stated in the text. What is the explanation for this fact?
Response: We appreciated the reviewer’s comment. The purpose of this study was to determine the optimal position and port placement during robotic resection for various mediastinal tumors. Although the position and approach for superior, posterior or middle mediastinal tumors do not differ much from previous reports, there are various approaches for anterior mediastinal tumors. Furthermore, it is not clear which approach is the best for anterior mediastinal tumors from previous reports. We presented our own experience of anterior mediastinal tumor resections because we thought it might be helpful to know which approach would be best for anterior mediastinal tumors. However, we can add our experiences of other mediastinal tumors if recommended by the editors and reviewers.
- It would be desirable to identify in the manuscript the most representative studies on which the review has been based and reflect them in the legend of the text. The use of an accessory table would be appreciated
Response: We appreciated the reviewer’s comment. We added a table (Table 2), (line 234-237).
- It is not usual to include a summary table with the personal recommendations of the authors in the conclusions of a review. Please consider its inclusion in a different section of the manuscript.
Response: We appreciated the reviewer’s comment. We made a new section and moved Table 2 to the section 6 “Literature summary and preferred position and approach for RATS mediastinal tumor resection”.
- References are updated.
Response: We appreciated the reviewer’s comment. We updated and added references.
We thank the reviewer for the time and effort to help us to improve the manuscript.
Round 2
Reviewer 2 Report
The authors answered the raised questions appropriately.
Before it can be accepted for publication, I suggest English polishment for the newly added paragraph as well as the format adjustment (New table) in revised manuscript.
Congratulations!